# The impact of COVID-19 pandemic course in the number and severity of hospitalizations for other natural causes in a large urban center in Brazil

Luisa C. C. Brant [1]*, Pedro C. Pinheiro[1], Isis E. Machado [2], Paulo R. L. Correa[3], Mayara R. Santos [3], Antonio L. P. Ribeiro [1], Unaí Tupinambás[1], Christine F. Santiago[3], Maria de Fatima M. Souza [4], Deborah C. Malta[5], Valéria M. A. Passos [1]

1 School of Medicine, Universidade Federal de Minas Gerais, Belo Horizonte, MG, Brazil, 2 School of Medicine, Universidade Federal de Ouro Preto, Ouro Preto, MG, Brazil, 3 Municipal Health Secretariat, Belo Horizonte, MG, Brazil, 4 Vital Strategies, São Paulo, SP, Brazil, 5 Nursing School, Universidade Federal de Minas Gerais, Belo Horizonte, MG, Brazil

* luisabrant@gmail.com

**Data Availability Statement:** The data that support the findings of this study are publicly available at https://datasus.saude.gov.br/transferencia-de-

## Abstract

The COVID-19 pandemic may indirectly impact hospitalizations for other natural causes. Belo Horizonte is a city with 2.5 million inhabitants in Brazil, one of the most hardly-hit countries by the pandemic, where local authorities monitored hospitalizations daily to guide regulatory measures. In an ecological, time-series study, we investigated how the pandemic impacted the number and severity of public hospitalizations by other natural causes in the city, during 2020. We assessed the number and proportion of intensive care unit (ICU) admissions and in-hospital deaths for all-natural causes, COVID-19, non-COVID-19 natural causes, and four disease groups: infectious, respiratory, cardiovascular, and neoplasms. Observed data from epidemiological week (EW) 9 (first diagnosis of COVID-19) to EW 48, 2020, was compared to the mean for the same EW of 2015–2019 and differences were tested by Wilcoxon rank-sum test. The five-week moving averages of the studied variables in 2020 were compared to that of 2015–2019 to describe the influence of regulatory measures on the indicators. During the studied period, there was 54,722 hospitalizations by non-COVID-19 natural causes, representing a 28% decline compared to the previous five years (p<0.001). There was a concurrent significant increase in the proportion of ICU admissions and deaths. The greater reductions were simultaneous to the first social distancing decree or occurred in the peak of COVID-19 hospitalizations, suggesting different drivers. Hospitalizations by specific causes decreased significantly, with greater increase in ICU admissions and deaths for infectious, cardiovascular, and respiratory diseases than for neoplasms. While the first reduction may have resulted from avoidance of contact with healthcare facilities, the second reduction may represent competing causes for hospital beds with COVID-19 after reopening of activities. Health policies must include protocols to address hospitalizations by other causes during this or future pandemics, and a plan to face the rebound effect for elective deferred procedures.

arquivos/, and the steps to reach the data are described in a tutorial (S1 Text). Data was fully anonymized before access.

**Funding:** Dr Ribeiro is supported in part by CNPq (310679/2016-8 and 465518/2014-1), by FAPEMIG (PPM-00428-17 and RED-00081-16) and CAPES (88887.507149/2020-00). Deborah C Malta is partially financed by CNPq (CNPQ - 310177/2020-0). The project is financed by the Global Grants Program, Vital Strategies, São Paulo, SP, Brazil. The funders had no role in study design, data collection and analysis, decision to publish, or preparation of the manuscript.

**Competing interests:** The authors have declared that no competing interests exist.

## Introduction

The coronavirus disease 2019 (COVID-19) pandemic had direct and indirect impacts on health systems, with previous reports of hospitalizations declines in many countries, such as the US, Italy, and Denmark [1–3]. Brazil has been one of the most hardly-hit countries by the pandemic with more than 500,000 deaths reported by the end of June, 2021 [4], as well as reductions in hospitalizations by other non-COVID-19 natural causes [5, 6]. However, little is known about the changes in the profile of hospitalizations by natural non-COVID causes, including use of intensive care units (ICU) and in-hospital mortality, during the pandemic in low- and middle-income countries (LMIC) and an increase in the number of deaths at home attended by the Mobile Emergency Medical Service in Belo Horizonte has been observed [7].

The decline in hospital admissions by non-COVID-19 causes during the pandemic may result from different causes. First, the intended postponement of elective procedures to prevent a health system collapse due to the higher expected demands of hospitalizations by COVID-19 [1, 8]. Second, the avoidance of medical care by patients due to social distancing mandates or fear of contracting COVID-19 in healthcare facilities [9]. Third, competing causes for hospital beds in the peak of COVID-19 hospitalizations may also play a role in this reduction. In fact, a decline in hospital admission by urgent conditions, such as acute coronary syndromes and stroke have been reported [10], what may have led to excess mortality by these causes [11].

Understanding the drivers of the changes in hospitalization trends in a location are fundamental to provide insights to health authorities regarding what patients may have been under-treated and how to prepare for future pandemics or the continued COVID-19 threat, focusing in mitigating its effects on other medical conditions. In this context, we aimed to analyze the trends in hospitalizations in public hospitals, intensive care unit (ICU) admissions and in-hospital mortality by natural causes, including its main four groups: infectious, respiratory and cardiovascular diseases, and neoplasms, during the COVID-19 pandemic in 2020, in Belo Horizonte, Brazil. These data may contribute to understand the impact of the pandemic trajectory and the social distancing decrees issued by the municipal authorities in the hospitalization trends in Belo Horizonte.

## Methods

### Setting, population and data

This is an ecological, time-series study conducted in Belo Horizonte city, the 6th largest city in Brazil, with 2.5 million inhabitants, located in the Southeast region. In Brazil, the public health system (Unified Health System, *Sistema Único de Saúde*, SUS) is constitutionally designed to provide the health as a "right for all" and a "duty of the state" [12], and as such public hospitalizations are free of charge for all Brazilian citizens, although 50% of Belo Horizonte's residents–usually those with higher socioeconomic conditions–have private health insurances and use private hospitals [13]. In Belo Horizonte, the public health system infrastructure is well organized compared to the rest of the country–the city has one of the best quality primary health care program [14], and an organized mobile medical services, though inequalities in access to care cannot be excluded [7].

During the COVID-19 pandemics, Belo Horizonte's municipality composed a committee of specialists to address the pandemics issues and help local authorities to decide about the timely implementation of regulatory measures, following three daily indicators: transmission rate, and the proportion of occupied hospital and ICU beds. Through this surveillance, the municipality was able to prevent an early hospital collapse due to adaptation of the health

system, creating new hospital and ICU beds as can be seen in **S1 Fig**. Other actions taken by the municipality to confront the pandemic have been published [15].

We analyzed data from the Hospital Information System (*Sistema de Internações Hospitalares, SIH*, in Portuguese) of the Brazilian Unified Health System (*Sistema Único de Saúde, SUS*, in Portuguese), provided by Belo Horizonte's Health Department (*Secretaria Municipal de Sáude, SMS-BH*, in Portuguese), after ethical approval. The SIH database is an administrative database for public hospitalizations, which is also used for hospital payment, resulting in no underreporting. The dataset included all public hospitalizations of the municipality and we extracted data for residents of Belo Horizonte hospitalized for natural causes from January, 2015 to November, 2020 (epidemiological weeks, EW, 1 to 48) in March, 2021. Therefore, we excluded hospitalizations for external causes (ICD10 S00-Y98), child-birth (O00-P96), and factors influencing health status and contact with services (ICD10 Z00-Z99). The decision to exclude the last EW of the years studied was based in the date of data extraction, because previous quality checks of the data revealed that on average 90% of hospitalizations were processed within a 3 months-period after the date of hospitalization. This reporting delay between hospitalization and processing month can be seen in **S1 Table**.

## Definition of causes and measures

We examined by EW the following variables: number of hospitalizations, number and percent of hospitalizations resulting in ICU admissions, or deaths (in-hospital deaths), for all natural causes (including COVID-19), COVID-19, and non-COVID-19 (non-COVID-19 natural causes). Hospitalizations were considered by COVID-19 if the B34.2 ICD-10 code were used. In quality control analysis, this code has not been identified in hospitalizations from the previous 5 years. COVID-19 ICD-10 code could only be assigned to hospitalizations in which patients had a positive diagnostic test, either RT-PCR or serology, that should have been requested to symptomatic patients upon hospitalization [16, 17]. The presence of a positive COVID-19 test was checked by auditors to avoid miscoding, particularly because COVID-19 hospitalizations paid more than other clinical hospitalizations. As each hospitalization in SIH is assigned only one code, if a positive test was present, a COVID-19 code was assigned, irrespective of comorbidities. Importantly, although there was a shortage of COVID-19 diagnostic tests in Belo Horizonte's community setting, these tests were widely available in hospitals.

We then further classified the non-COVID-19 hospitalizations according to the following ICD-10 chapters, which includes the diseases that are the leading natural causes of hospitalizations in Brazil and Belo Horizonte [18]: I) Certain infectious and parasitic diseases, II) Neoplasms, IX) Diseases of the circulatory system, and X) Diseases of the respiratory system. The duration of hospitalizations were also analyzed for all groups of causes. We have also described the profile of hospitalizations according to the pandemic phase in the city. The first COVID-19 diagnosis in Belo Horizonte occurred in EW 9, and the dates of the most important health policies in the pandemic course in the city were marked in the figures: (I) dispatch of a municipal decree implementing the first control phase (EW 11), (II) first gradual reopening of business facilities (EW 21), (III) beginning of second control phase (EW 26), (IV) beginning of second gradual reopening (EW 31). During the control phase, non-essential business facilities and schools were closed and essential business had time restrictions. Importantly, schools were not included in the gradual opening in 2020 [7].

## Statistical analysis

We investigated the profile of hospitalizations by sex and age group: young (0–29 years old), adults (30–59 years old), and older adults (60+ years old), for the different groups of

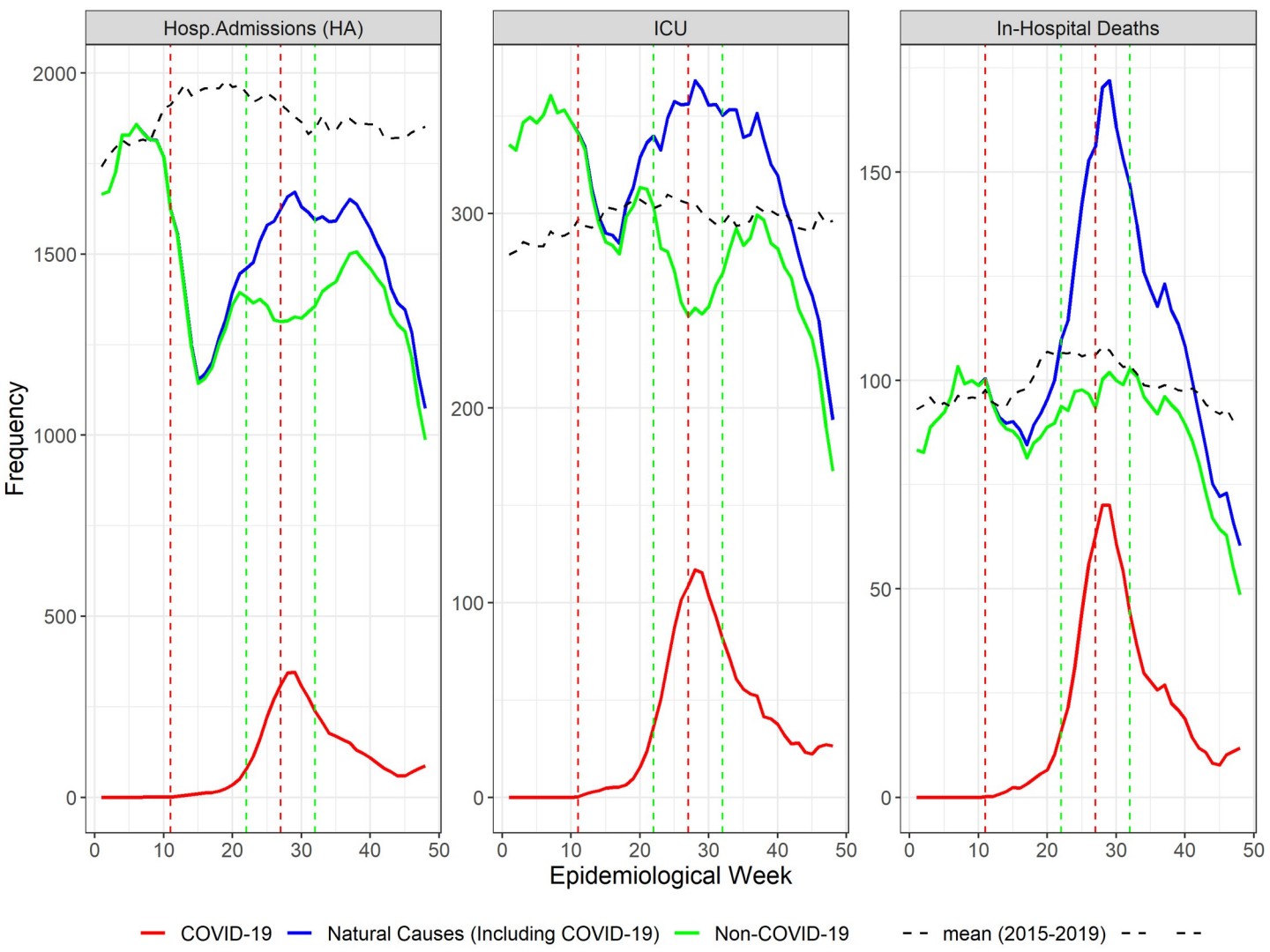

**Fig 1. Five-week moving average of number of hospital admissions, intensive care unit admissions, and in-hospital deaths in Belo Horizonte, epidemiological weeks (EW) 1 to 48, 2020 (solid lines), and the mean of the same EW of 2015–2019 (dashed line) for: All natural causes (green), non-COVID-19 natural causes (blue), and COVID-19 (red).** Vertical lines refer to the weeks of control (red) and reopening phases (green).

hospitalization causes. Proportions of ICU admissions and deaths were defined as the ratio between the number registered for each variable and the total number of hospitalizations, considering each age group, sex and cause of hospital admission. In **Figs 1–4**, the five-week moving averages of the studied variables from January to November, 2020, were compared to a single estimate: the five-week moving averages of the same variables for the previous period (January 2015 to November 2019), for each group of diseases. The lower and upper values for the same EW from 2015 to 2019 were also included in all tables and figures. To investigate trends in hospitalizations and ICU admissions along the years 2015 to 2020, we graphically compared observed data from EW of each cited year to the mean, lower and upper limits of 2015–2019. In **Table 1**, the unsmoothed observed variables from EW 9 to 48 of 2020 were compared to their unsmoothed mean for the same EW from the 2015–2019 period. For

frequencies (hospitalizations, ICU admissions, and in-hospital deaths), we present the sum for the EW 9–48 period, and the respective means for proportions (% ICU admission, % in-hospital deaths). For all variables, the difference between the observed data from EW 9–48 of 2020 and the 2015–2019 mean was tested using the Wilcoxon rank sum test, as the distributions were not normal. We compared 2020 estimates with a single estimate: the observed mean in the 2015–2019 period. The resulting P-values were afterwards corrected using the Benjamini & Hochberg correction for multiple testing and were considered significant if $p \leq 0.05$. In this analysis, we considered the start of the pandemic data from EW 9 because it was the week of the first diagnosis of COVID-19 in Belo Horizonte. Analyses were performed using R, version 4.0.1 (The R Foundation) [19].

## Ethical considerations

The study was approved by the Universidade Federal de Minas Gerais (UFMG) and SMS-BH Review Board (Protocol: CAAE 39778720.4.3001.5140). We followed the STROBE guidelines for reporting observational studies. The data that support the findings of this study are publicly available at https://datasus.saude.gov.br/transferencia-de-arquivos/, and the steps to reach the data are described in a tutorial (**S1 Text**). Data was fully anonymized before access.

## Results

In the evaluated period of 2020, there were 59,397 hospitalizations for all natural causes (including COVID-19) in Belo Horizonte. For non-COVID-19 natural causes, there were 54,722 hospitalizations, representing a 28% significant reduction, when compared to the mean of the previous five years. This substantial reduction in hospitalizations occurred with a concurrent maintenance in the duration of hospital stays for non-COVID-19 causes (**S2 Table**).

**Table 1** reveals that there was a decline for hospitalizations for non-COVID-19 natural causes and for the 4 selected groups of diseases. This reduction occurred concurrently with a reduction in the number of ICU admissions and in-hospital deaths, mainly driven by CVD and neoplasm. However, the proportion of ICU admissions increased in the same period (+4.7%), as did the proportion of in-hospital deaths (+1.2%), except for neoplasms.

Analyses stratified by sex (**S3 Table**) did not reveal differences between sexes from what has been described for hospitalizations and ICU admissions. However the number of in-hospital deaths for CVD only decreased for women. Moreover, the proportion of in-hospital deaths for CVD and respiratory diseases only increased for women, while for neoplasms it only decreased for men. **S4 Table** shows reductions in hospitalization for non-COVID-19 and specific causes in all-age groups, though non-significant for respiratory diseases in older adults. Regarding ICU admissions for non-COVID-19 causes, a decrease occurred in all age groups, though only significant for young adults; while the proportion of ICU admissions was significantly higher for all age groups. For in-hospital deaths a significant reduction occurred for those adults with 0–29 y and 30-59y, and was non-significant for older adults, with consistent reductions for all age groups for CVD and neoplasms. For the proportion of in-hospital deaths, the changes by diseases groups were mostly non-significant.

**Figs 1–4** represent the trends in the profile of hospitalizations along 2020, and includes the control and reopening phases. **Fig 1** depicts a reduction in the number of non-COVID-19 hospitalizations, ICU admissions, and in-hospital deaths during the first control phase (EW 11–21). In the same period, a slow rise in the same indicators for COVID-19 is observed, which then rises steeply after the first reopening at EW 21 up to the implementation of the second control phase in EW 26. In the second control phase (EW 26–31), the decrease in hospitalizations for non-COVID-19 causes is smaller than in the first control phase. However, the

**Table 1. Difference in the number of hospitalization, and number and proportion of intensive care unit admission and in-hospital deaths between from epidemiological weeks 9–48, 2020 (observed), and the 2015–2019 mean for the same EW, in Belo Horizonte.**

| Variables | 2020 | 2015–2019[1] | Difference | P-Value |
|---|---|---|---|---|
| **Hospitalizations[2]** | | | | |
| Non-COVID-19 natural causes | 54722 | 75840 (68158;85370) | -21118 | <0.01 |
| Infectious diseases | 5599 | 6721 (5006;9603) | -1122 | <0.01 |
| Neoplasms | 8393 | 10890 (9721;12039) | -2497 | <0.01 |
| Cardiovascular diseases | 9459 | 12438 (11029;13766) | -2979 | <0.01 |
| Respiratory diseases | 7357 | 9804 (8401;11315) | -2447 | <0.01 |
| **ICU admissions[2]** | | | | |
| Non-COVID-19 natural causes | 11168 | 12011 (9859;14804) | -843 | 0.03 |
| Infectious diseases | 2223 | 2038 (1488;2751) | 185 | 0.10 |
| Neoplasms | 1245 | 1458 (1135;1798) | -213 | <0.01 |
| Cardiovascular diseases | 3601 | 4073 (3207;5016) | -472 | <0.01 |
| Respiratory diseases | 1276 | 1231 (830;1688) | 45 | 0.80 |
| **% ICU admissions[3]** | | | | |
| Non-COVID-19 natural causes | 20.5 | 15.8 (13.8;18.1) | 4.7 | <0.01 |
| Infectious diseases | 39.3 | 31.4 (23.7;38.9) | 7.9 | <0.01 |
| Neoplasms | 14.7 | 13.4 (10.9;16.0) | 1.3 | 0.04 |
| Cardiovascular diseases | 38.1 | 32.9 (25.9;39.5) | 5.2 | <0.01 |
| Respiratory diseases | 17.7 | 12.7 (9.0;16.6) | 5.0 | <0.01 |
| **In-Hospital deaths[2]** | | | | |
| Non-COVID-19 natural causes | 3561 | 4003 (3444;4687) | -442 | <0.01 |
| Infectious diseases | 1293 | 1170 (861;1537) | 123 | 0.02 |
| Neoplasms | 600 | 902 (661;1152) | -302 | <0.01 |
| Cardiovascular diseases | 588 | 669 (467;903) | -81 | <0.01 |
| Respiratory diseases | 453 | 480 (304;688) | -27 | 0.30 |
| **% In-Hospital deaths[3]** | | | | |
| Non-COVID-19 natural causes | 6.5 | 5.3 (4.6;6.0) | 1.2 | <0.01 |
| Infectious diseases | 2.3 | 18.3 (13.2;23.4) | 4.7 | <0.01 |
| Neoplasms | 7.1 | 8.4 (6.0;10.9) | -1.3 | <0.01 |
| Cardiovascular diseases | 6.2 | 5.4 (3.7;7.1) | 0.8 | <0.01 |
| Respiratory diseases | 6.2 | 5.0 (3.2;6.9) | 1.2 | <0.01 |

[1]2015–2019 mean. In parenthesis, lower limit aggregates the lowest values for the same EW between 2015–2019 and the upper limit the aggregates the highest.

[2]Sum of the observed values EW 9–48.

[3]Mean of the observed values EW 9–48.

number of ICU admissions decreased more steeply, and in-hospital deaths increased to values close to the in-hospital mortality of previous years, even in a scenario with significantly less hospitalizations. Of note, the peak of hospitalizations, ICU admissions and in-hospital deaths due to COVID-19 occurs in parallel with a reduction in hospitalizations and ICU admissions, and an increase in in-hospital mortality for non-COVID-19 causes.

**Fig 2** represents the number of hospitalizations in 2020, compared to the mean of hospitalizations in the previous five years. Just before EW11, there was a consistent decrease in hospital admissions for neoplasm, cardiovascular and respiratory diseases, which although more evident during the first control phase (EW 11–21), does not return to previous numbers until the end of the analysed period–except for respiratory diseases for which it reaches the lower values after the second control period (EW26 onwards). Of note, there is a steep decline for respiratory diseases in the first control phase, which coincides with the seasonal influenza period. The

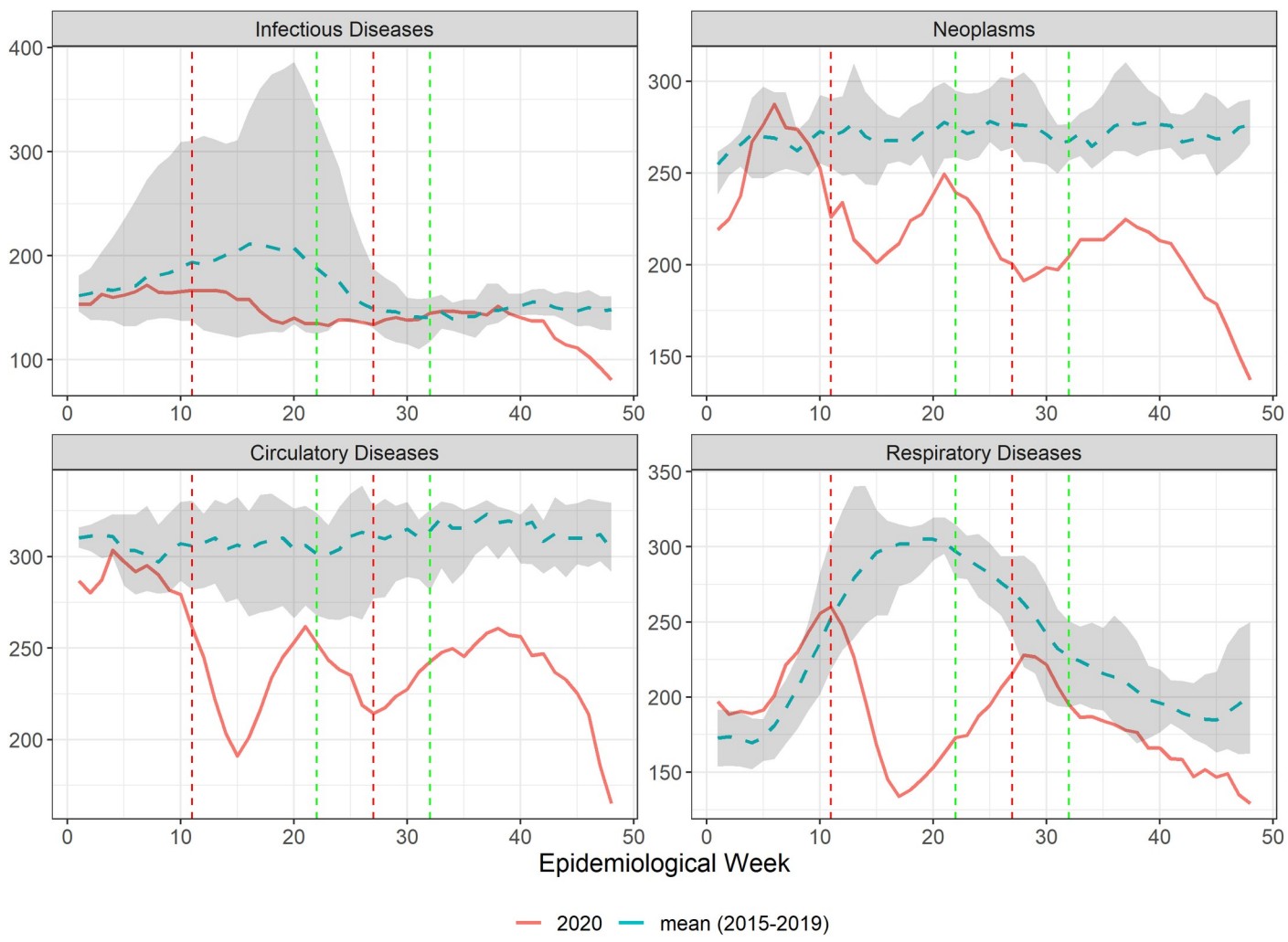

Red dashed vertical lines: Control Phase Established;
Green dashed vertical lines: Reopening;

Grey Ribbon indicates the maximum and minimum observed (moving average) between 2015 and 2019

**Fig 2. Five-week moving average of number of hospital admissions in Belo Horizonte during the COVID-19 pandemic from epidemiological weeks (EW) 1–48, 2020 (solid line), compared to the mean of 2015–2019 (dashed line), for the 4 selected disease groups.** Vertical lines refer to the weeks of control (red) and reopening phases (green).

number of hospitalizations for infectious diseases remained near the lowest values of the five previous years, even during the period of seasonality.

In **Fig 3**, we observed that the number of ICU admissions for all disease groups in 2020 were similar to the previous five years—considering the interval represented by the lowest and highest number of ICU admission during 2015 to 2019—except for neoplasms and cardiovascular diseases that, during the COVID-19 peak in the city (second control phase), when a reduction in ICU admissions was observed. On the other hand, the proportion of ICU admissions was higher during the first control phase (EW11-EW21) for the four diseases groups and after EW 30 for infectious and respiratory diseases. Interestingly, there were higher numbers and proportion of ICU utilization in the beginning of 2020, before the COVID-19 spread in the city. In fact, **S2 and S3 Figs** reveal that the number of hospitalizations and ICU admissions

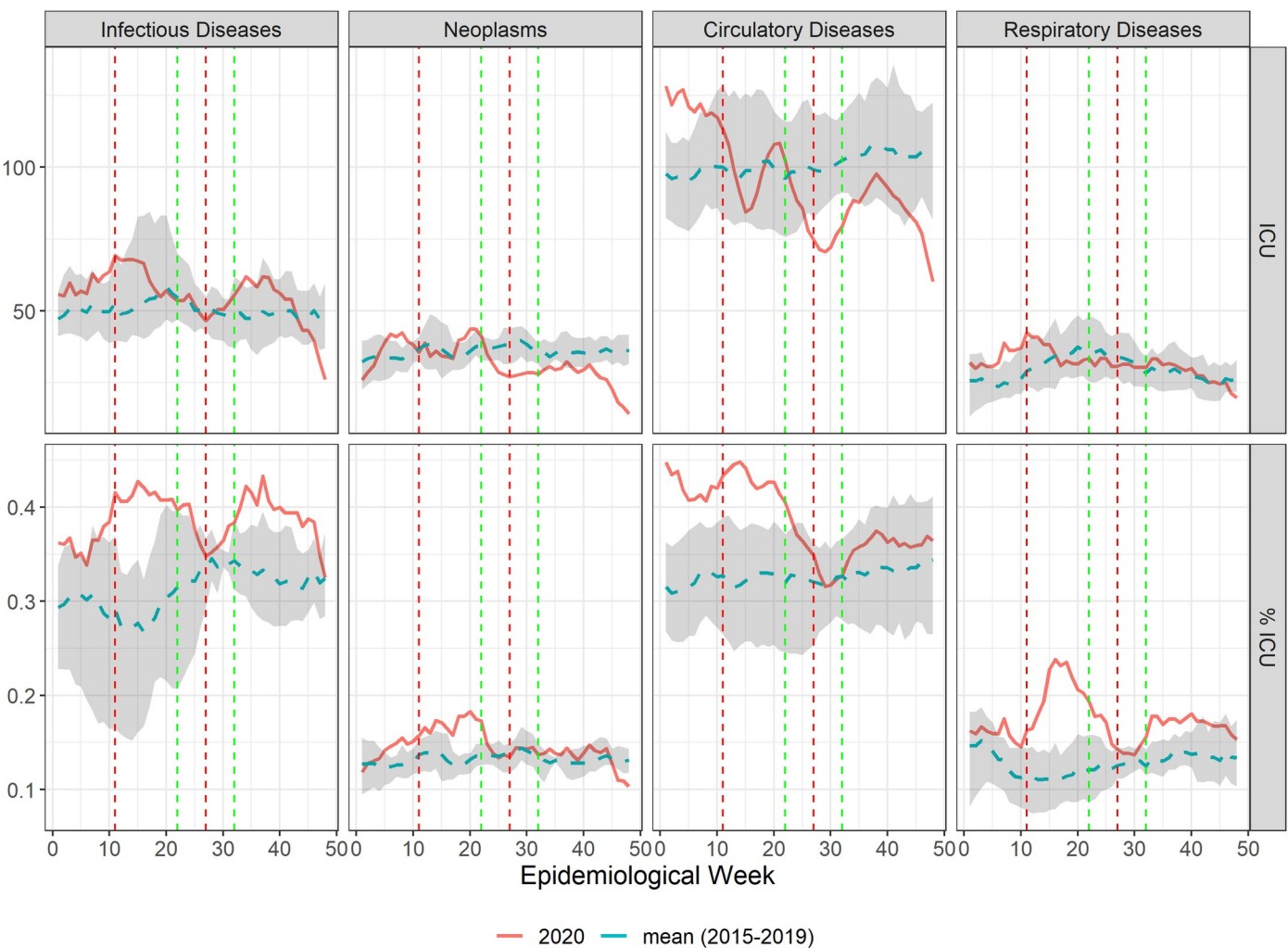

Red dashed vertical lines: Control Phase Established;
Green dashed vertical lines: Reopening;

Grey Ribbon indicates the maximum and minimum observed (moving average) between 2015 and 2019

**Fig 3. Five-week moving average of number of Intensive Care Unit admissions in Belo Horizonte during the COVID-19 pandemic from epidemiological weeks (EW) 1–48, 2020 (solid line), compared to the mean of 2015–2019 (dashed line), for the 4 selected disease groups.** Vertical lines refer to the weeks of control (red) and reopening phases (green).

were progressively rising in the city from 2015 to 2019, with a change in this rising trend to a decline when the pandemics started.

For in-hospital mortality (**Fig 4**), a heterogeneous profile was observed for each group of diseases. There was an increase in both the number and proportion of deaths for infectious diseases from EW21 to EW40, in parallel with the peak of COVID-19, and the proportion of deaths due to respiratory disease was slightly higher from EW10 to EW35. There was no change in the proportion of in-hospital mortality by CVD, although there was a slight reduction in the number of deaths, during the periods of hospitalizations' decline. For neoplasm, while the proportion of deaths remained the same, smaller number of deaths was observed along the 2020 pandemics period. Importantly, because the numbers of weekly in-hospital deaths are relatively low, changes have to be interpreted with caution.

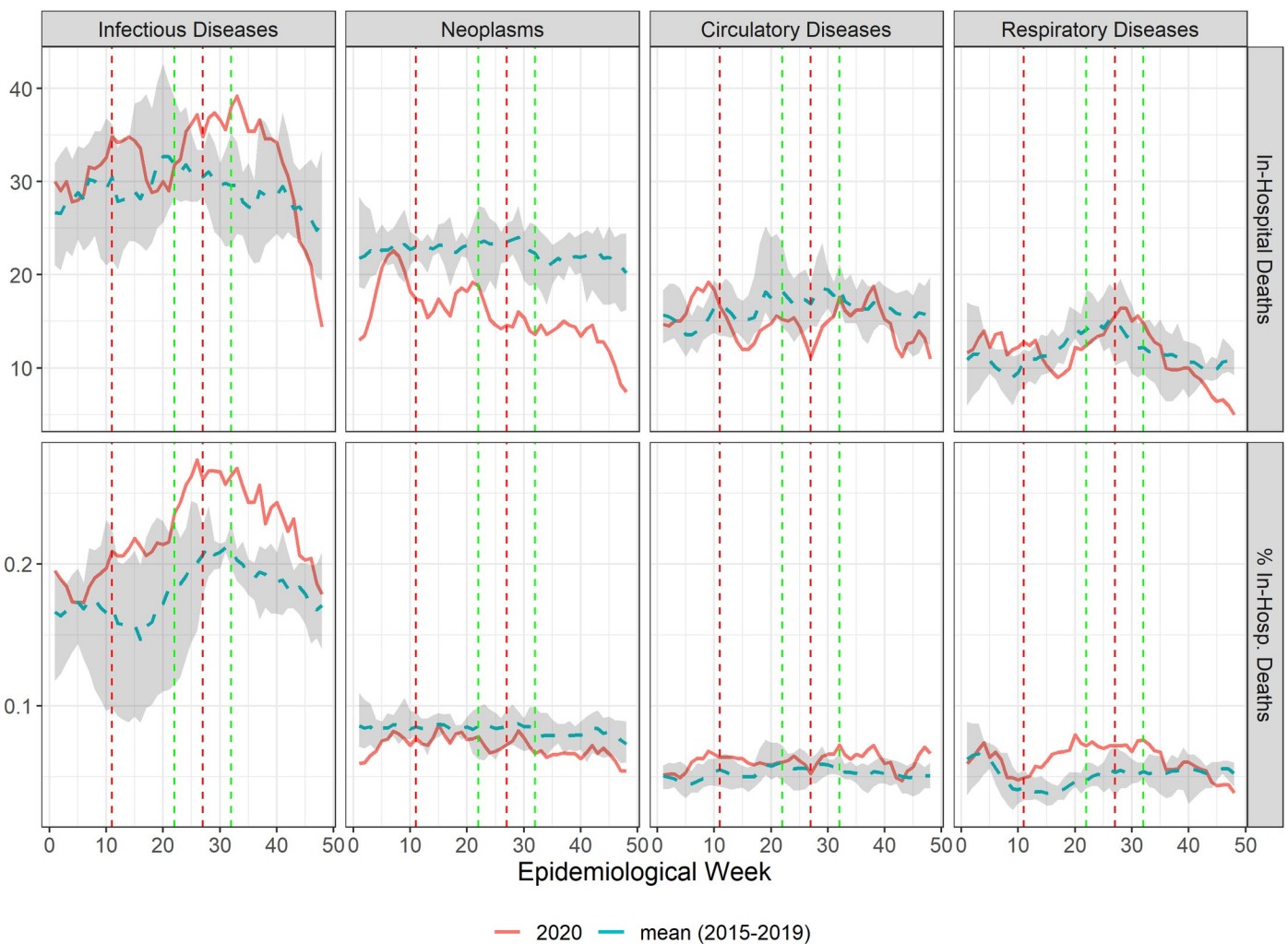

2020 mean (2015-2019)

Red dashed vertical lines: Control Phase Established;
Green dashed vertical lines: Reopening;

Grey Ribbon indicates the maximum and minimum observed (moving average) between 2015 and 2019

**Fig 4. Five-week moving average of number of in-hospital mortality in Belo Horizonte during the COVID-19 pandemic from epidemiological weeks (EW) 1–48, 2020 (solid line), compared to the mean of 2015–2019 (dashed line), for the 4 selected disease groups.** Vertical lines refer to the weeks of control (red) and reopening phases (green).

## Discussion

The indirect impact of the COVID-19 pandemic in hospitalizations in Belo Horizonte during 2020 follows the same trend already described in other countries and Brazilian cities [1–3, 5, 6] there was a reduction in hospital admissions by non-COVID-19 natural causes, when compared to the 5 previous years, which did not return to baseline levels until the end of 2020. Importantly, we broadened the findings demonstrating that in parallel with the reduction in hospital admissions, there was an increase in the proportion of non-COVID-19 hospitalizations that used intensive care or had in-hospital mortality as outcome, both findings suggesting either hospitalizations of more severely ill individuals or delivery of lower quality/delayed care to non-COVID-19 patients during the pandemic. In fact, previous analysis in other location has shown that the reduction in emergency department visits was greater for less severe cases

[2] and that acute conditions with time-sensitive treatment, such as acute myocardial infarction, received delayed treatment [20].

The reduction in the number of hospitalizations by non-COVID-19 natural causes during the pandemic has been generally described as having multifactorial causes, from changes in populations' behaviour to health system reorganization, with the deferment of elective procedures, or collapse [1, 3]. However, how these components interact in the course of the pandemic has been less explored. Looking at the pattern of non-COVID-19 hospitalizations along the studied period, we can clearly see two peaks of consistent reductions in 2020 that may have been led by different drivers. The first peak occurred after the primary diagnosis of COVID-19 in Belo Horizonte and the dispatch of a municipal decree that closed non-essential business and schools. Apparently, it resulted from a change in the behaviour of the population, who may have deferred care following recommendations of health authorities—who were threatened by possible high hospital demands—or due to perceived risks of contracting COVID-19. This hypothesis is corroborated by the concomitant decline of ICU admissions and in-hospital deaths in the period, and also by previous studies that depicted a fall in health services utilization in Brazil during the pandemic [9]. On the other hand, during the second decline in non-COVID-19 admissions, which started after a partial recovery in the number of hospitalizations, there was a reduction in ICU admissions for non-COVID-19 causes not accompanied by a reduction in in-hospital mortality. The second peak of decline occurred in parallel with the rise in COVID-19 hospitalizations in the city and, as such, may have resulted from the additional effect of hospital capacity constraints, which could also have contributed to higher in-hospital mortality.

Conversely, it is worth mentioning that fewer hospitalizations could be a consequence of lower incidences of the diseases that cause hospital admissions, possibly as a result of healthy changes in lifestyle, lower air pollution or less transmission of communicable diseases due to social distancing, mask use and hygiene measures [21, 22]. While this in fact may be true for the latter, resulting in lower incidences of respiratory communicable diseases, previous studies in Brazil with 45,000 participants have shown that unhealthy behaviours, such as smoking, alcohol use and a sedentary lifestyle, have actually increased during the pandemic [23–25]. Furthermore, hospital admissions may decline due to the enhancement of modalities of care that may prevent hospitalization, such as homecare and telemedicine [26, 27]. However, the population included in our analysis, who uses public hospitals in Brazil, have negligible access to the cited alternative modalities of care. Longer hospitalizations may also reduce the number of admissions, however duration of hospitalizations for non-COVID causes did not change in the pandemics period. We hypothesized that this could be the result of two opposite trends: greater clinical severity, but an urge from clinicians to discharge patients to minimize risk of contracting COVID-19. As such, neither a reduction in the incidence of diseases, changed treatment pathways, nor longer hospitalizations seem to be alternative causes for the decline in hospitalizations herein reported.

The analysis stratified by sex did not reveal consistent significant differences, with the exception of the higher proportional in-hospital mortality by CVD and respiratory diseases for women. As such, the COVID-19 pandemic may be further increasing the gap in treatment for cardiovascular diseases in women [28], although due to the limits of data after stratification, these findings must be interpreted with caution. Regarding the analysis by age groups, the pattern of hospitalization changes during the pandemic was similar and although no additional insights for non-COVID-19 hospitalization was provided, it is important to emphasize how COVID-19 had greater impact in older adults.

Finally, the pattern of hospitalizations observed for each group of disease is diverse, revealing that the pandemic differentially disrupts routine care according to the characteristics of the

diseases. For infectious disease, there was a decline in the number of hospitalizations during the first control phase that returned to baseline levels, enhanced by the higher number of hospital admissions during the period in previous years caused by dengue epidemics in the city in 2016 and 2019 [29]. However, the reduction in hospitalizations occurred in parallel with a proportional rise in ICU admissions, suggesting hospitalization of more severely ill patients, impaired delivery of care, or previous trends halted by the pandemics [30]. Interestingly, during and shortly after the peak of COVID-19 hospitalizations (second control phase), there was a rise in in-hospital deaths, probably as a result of compromised care. Respiratory diseases had the steepest drop in the number of hospitalizations during the first control phase, which matches the seasonal influenza period, suggesting that social distancing, mask use and enhanced hygiene measures may have lowered influenza transmission [22, 31, 32]. Regarding ICU use and in-hospital deaths, we observed a rise in the proportion of ICU utilization and in-hospital deaths during the first control phase, suggesting the same drivers already described for infectious diseases.

Regarding non-communicable diseases–cardiovascular and neoplasms–, the declines in hospital admissions follow the same temporal patterns, with two peaks in both control phases, as described. However, the proportion of ICU admission is higher along the first control, but then declines in the peak of COVID-19 hospitalizations, revealing hospital capacity collapse at this point. Interestingly, while the proportion of in-hospital deaths for cardiovascular diseases were higher than in previous years, the opposite was detected for neoplasms, suggesting that individuals with cancer may have died at home, probably due to reduced hospitalizations for palliative care.

The limitations of the present analysis include the lack of comprehensiveness of the data source used, which does not include all hospitalizations in the municipality but rather those of the Brazilian public health system (*Sistema Único de Saúde*, SUS). Although the SUS is a universal health system, 50% of Belo Horizonte's inhabitants have health insurance, and as such may not use public hospitals [33]. While we acknowledge this is a limitation, we are studying the impact of the pandemic in the inhabitants of Belo Horizonte of lower socioeconomic status, depicting the negative effect of the pandemic in this group. Moreover, as this is the population mostly affected by health policies related to public healthcare facilities, understanding the changes in patterns of hospitalizations in public hospitals are crucial for ensuring preparedness to specific diseases during the continued threat of COVID-19 and future pandemics, focused to this group.

Another issue regarding the data source is its completeness and miscoding. To overcome that, we did a thorough quality analysis and excluded the last 5 EW of 2020, when the loss in the number of hospitalizations could not be considered negligible in previous years, being the quality analysis one of the main strengths of our study. However, due to the changes in routines during pandemic times, we cannot assure the pattern was the same in the end of 2020, and the decline in hospitalizations from EW 45 onwards may result from reporting delays. About miscoding, first, coding procedures for non-COVID-19 hospitalizations–the focus of our analysis–did not change during the period. Because COVID-19 could only be coded in the presence of a positive diagnostic test, the main potential bias would be to have hospitalized patients with COVID-19 that would receive an alternative diagnosis in case a COVID-19 diagnostic test was not requested. In this scenario, the real decline in non-COVID-19 hospitalizations would be even greater, reinforcing our results. Other strengths of our study include the evaluation of the pandemic trajectory in the city, and the detailed analysis of ICU use and in-hospital deaths by group of causes, providing information for local authorities on how to address the potential determinants of the changes herein reported.

Taken together, our analysis shows that the COVID-19 pandemic resulted in a decline in hospitalizations by non-COVID-19 natural causes in Belo Horizonte in 2020. The greater reductions were preceded by the first social distancing decree or occurred in the peak of COVID-19 hospitalizations in the city, suggesting different drivers. While the first greater reduction in hospitalizations by natural causes may have resulted from fear of contracting COVID-19 or adherence to social distancing measures, the second reduction may have had a greater effect of competing causes for hospital beds with COVID-19, resulting in a reduction in the proportion of ICU utilization and higher in-hospital mortality by non-COVID causes.

The potential deleterious consequences of a decline in hospital admissions for non-COVID-19 natural causes during the pandemic are an increase in overall mortality due to reduced access to care and greater burden of prevalent diseases due to the postponement of elective procedures [3]. As such, beyond preparing the system to deal with the rebound effect of deferred procedures in the near future, public health authorities will need to address different scenarios to mitigate the indirect effects of novel waves of the present pandemic or others that may occur. Inform the population how to act in the presence of medical conditions that require urgent management–including reorganizing pathways of care–and prevent the health system collapse, which increased in-hospital deaths by COVID-19 and other causes. In Brazil, the lack of coordination for tackling the pandemic from federal authorities may have contributed to the unfavourable patterns of hospitalizations shown. Instead of prioritizing population-wide strategies, as the dissemination of scientific based information to educate the population and the promotion of mass testing and then vaccination to address the pandemic, indeed the federal government emphasizes the use of drugs of unproved efficacy for the "early treatment" of COVID-19 [34]. Even though Belo Horizonte's local authorities daily monitor the transmission rate, and the number of hospital and ICU admissions to define regulatory measures, among other actions [11], a national unified effort to reduce SARS-COV2 transmission and prepare the health system has been shown to be fundamental.

## Supporting information

**S1 Fig. Number of hospital and intensive care unit beds from January, 2015 to December, 2020, in Belo Horizonte.**
(PNG)

**S2 Fig. Five-week moving average of number of hospital admissions observed in Belo Horizonte in the epidemiological weeks (EW) 1 to 48, from 2015 to 2020 (blue lines), and the mean of the same EW of 2015–2019 (red lines) for all natural causes.** Vertical lines refer to the weeks of control (red) and reopening phases (green). Grey ribbon indicates the maximum and minimum observer (moving average) for the period.
(TIF)

**S3 Fig. Five-week moving average of number of intensive care unit admissions observed in Belo Horizonte in the epidemiological weeks (EW) 1 to 48, from 2015 to 2020 (blue lines), and the mean of the same EW of 2015–2019 (red lines) for all natural causes.** Vertical lines refer to the weeks of control (red) and reopening phases (green). Grey ribbon indicates the maximum and minimum observer (moving average) for the period.
(TIF)

**S1 Table. Hospital admission month and hospitalization processing month in Belo Horizonte, from January 2019 to May 2020.**
(DOCX)

**S2 Table. Mean and standard deviation (in parentheses) of the duration of hospital stay in days for selected groups of diseases, in Belo Horizonte, from 2015 to 2019 and in epidemiological weeks 9–48 of 2020.**
(DOCX)

**S3 Table. Difference in the number of hospitalization, and number and proportion of intensive care unit admission and in-hospital deaths from epidemiological weeks (EW) 9–48, 2020 (observed), and the 2015–2019 mean for the same EW, according to sex, in Belo Horizonte.**
(DOCX)

**S4 Table. Difference in the number of hospitalization, and number and proportion of intensive care unit admission and in-hospital deaths from epidemiological weeks (EW) 9–48, 2020 (observed), and the 2015–2019 mean for the same EW, according to age groups, in Belo Horizonte.**
(DOCX)

**S1 Text. Tutorial to access DATASUS data used in the present analysis.**
(DOCX)

## Author Contributions

**Conceptualization:** Luisa C. C. Brant, Isis E. Machado, Paulo R. L. Correa, Mayara R. Santos, Antonio L. P. Ribeiro, Maria de Fatima M. Souza, Deborah C. Malta, Valéria M. A. Passos.

**Data curation:** Paulo R. L. Correa, Mayara R. Santos.

**Formal analysis:** Pedro C. Pinheiro.

**Funding acquisition:** Luisa C. C. Brant, Isis E. Machado, Paulo R. L. Correa, Mayara R. Santos, Antonio L. P. Ribeiro, Maria de Fatima M. Souza, Deborah C. Malta, Valéria M. A. Passos.

**Methodology:** Luisa C. C. Brant, Pedro C. Pinheiro, Isis E. Machado, Paulo R. L. Correa, Mayara R. Santos, Antonio L. P. Ribeiro, Unaí Tupinambás, Christine F. Santiago, Maria de Fatima M. Souza, Deborah C. Malta, Valéria M. A. Passos.

**Supervision:** Luisa C. C. Brant, Maria de Fatima M. Souza, Deborah C. Malta, Valéria M. A. Passos.

**Visualization:** Pedro C. Pinheiro, Isis E. Machado.

**Writing – original draft:** Luisa C. C. Brant, Valéria M. A. Passos.

**Writing – review & editing:** Pedro C. Pinheiro, Isis E. Machado, Paulo R. L. Correa, Mayara R. Santos, Antonio L. P. Ribeiro, Unaí Tupinambás, Christine F. Santiago, Maria de Fatima M. Souza, Deborah C. Malta.

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
