## [Decision Letter · Decision Letter 0]

23 Aug 2021

 PGPH-D-21-00228 The impact of COVID-19 pandemic course in the number and severity of hospitalizations for other natural causes in a large urban center in Brazil PLOS Global Public Health

Dear Dr. Brant,

Thank you for submitting your manuscript to PLOS Global Public Health. After careful consideration, we feel that it has merit but does not fully meet PLOS Global Public Health’s publication criteria as it currently stands. Therefore, we invite you to submit a revised version of the manuscript that addresses the points raised during the review process.

We look forward to receiving your revised manuscript.

Kind regards,

Mathieu Nacher

Academic Editor

Journal Requirements:

Additional Editor Comments (if provided):

Reviewers' comments:

Reviewer's Responses to Questions

**Comments to the Author**

1. Does this manuscript meet PLOS Global Public Health’s publication criteria? Is the manuscript technically sound, and do the data support the conclusions? The manuscript must describe methodologically and ethically rigorous research with conclusions that are appropriately drawn based on the data presented.

Reviewer #1: Partly

Reviewer #2: Partly

2. Has the statistical analysis been performed appropriately and rigorously?

Reviewer #1: No

Reviewer #2: I don't know

3. Have the authors made all data underlying the findings in their manuscript fully available (please refer to the Data Availability Statement at the start of the manuscript PDF file)?

Reviewer #1: Yes

Reviewer #2: No

4. Is the manuscript presented in an intelligible fashion and written in standard English?

Reviewer #1: No

Reviewer #2: Yes

5. Review Comments to the Author

Reviewer #1: Conclusions are partly supported by the data. The aim of this study is to highlight the COVID-19 pandemic impact upon the number and severity of hospitalisation in public hospital, by non-COVID 19 natural causes, in Belo Horizonte city in 2020. Nevertheless, some relevant data could be presented and discussed, to avoid possible confusion biases, as the number of bed available in public hospitals for non COVID-19 hospitalisations, duration of hospital stay in 2015-2019 versus 2020 and number of in-patients' COVID-19 tests carried out if any. More explanation about the efficiency of public health policies deployed during this 2020 pandemic episode should be of interest (i.e. Mobile Emergency Medical Service).

The economic characteristic of the population attending public health hospitals in Belo Horizonte city could be presented since introduction as a part of context. In the same way, the decrease of hospitalisation during slow-down measures could be due to a loss of income in a part of the population (day labourers / informal workers).

Data need to be presented in a more rigorous way : some discrepancies exist in the data presented in Results (lines 154-155) and Table 1 : hospitalisation for natural causes and hospitalisation for non-COVID-19 natural causes are respectively 59,397 and 54,722 versus 73,690 and 69,015.

Some sentences could be written in a more intelligible way (lines 60-64 ; 162-165 ; 279-282). Figures could be clarified : only one variable by figure, allowing comparison between ICD-10 categories and corresponding units added.

At the time of review, data were not available on the website http://www2.datasus.gov.br/DATASUS/index.php?area=0901&item=1&acao=25 are not available : "Erro no acesso à página".

Reviewer #2: Thank you for submitting this manuscript. The topic is of relevance to a public health audience and in particular to health policy makers. I have a few preliminary concerns which would need to be addressed before reaching a final conclusion about the main statements brought by the authors in the results and discussion sections.

1) My first concern is about the possible systematic underreporting of all hospital admissions, regardless of the cause, in the context of COVID19 epidemic. This is briefly addressed in the discussion by the authors but in my opinion this could be a major bias that affects the message of this work. Do the authors have access to another source of data that would help estimating this systematic underreporting?

2) My second concern is about a possible first wave of COVID19 from EW0 to EW15. How can the peak observed between EW0 and EW15, in particular in all causes ICU admissions, be explained? My understanding is that the first case of COVID19 in Belo Horizonte was reported in EW9, however can the possibility of a first wave of COVID19 between EW0 and EW15 be ruled out? Has this been investigated? What alternative hypotheses can be given to explain this rise in all-causes ICU admissions?

3) My third concern results from the second: if a first wave of COVID19 gone undetected is plausible then my conclusion is that there was a significant underreporting of cases of COVID19 until coding practices were well established (either because of increased professional awareness, improved availability of specific diagnosis, etc.). This seems to be the case only after EW20 when the trend of COVID19-related hospital and ICU admissions actually followed that of all-causes, after a delay of about one month (the steep rise in hospital admissions and ICU is visible from EW15-EW17). For this reason, I would only draw conclusions on the differences between COVID19 and non-COVID19 related causes of hospitalizations after EW20. Conclusions on all causes remain valid for the whole period.

4) Many statistical tests were conducted but I do not have enough information to conclude about their relevance, please refer to my comments for the Methods and Results sections. The main message of the manuscript would benefit from a more concise reporting of the results, especially for the four categories of non-covid causes of admissions (Table 1 and Figures 2-5).

Introduction

The introduction is clear and concise, the context is well described, the objectives and underlying hypotheses well exposed. The topic is indeed highly relevant. I have no specific comments about this section.

Methods

More information would be welcome regarding how patients were classified in Belo Horizonte with the ICD coding. Could a patient be classified with several different ICD codes, if so, which one was chosen and according to which rule? For example would a patient with cardiac failure aggravated by COVID19 admitted in cardiology ICU by classified?

Was the coding procedure consistent? How were these practices impacted by the epidemic? Did they improve over time?

How were cases of COVID19 identified in Belo Horizonte in 2020? What diagnostic tests were available and when? Was the case definition consistent over time?

Line 103: Could the authors explain how this quality checks were performed for the previous years (how was the 90% figure determined, with what reference data) and whether this was also performed for 2020? This could help give insight on a possible systematic underreporting in 2020.

Line 129: Are ICU admissions included in the total hospitalization admissions?

Lines 138-140: Wilcoxon rank sum test was used. Can the authors confirm the following: for each outcome variable the statistical test compared a single estimate (for year 2020) to the distribution of five estimates (for each year 2015 to 2019)?

Line 141: Referring to my concern of a possible unreported wave of COVID19 between EW0 and EW15 and possible associated lack of appropriate coding of COVID19 cases: I would suggest to begin the analysis at EW0 for all natural causes, or EW20 when distinguishing COVID19-related and non-COVID19-related causes.

Line 142: How was the Benjamini & Hochberg correction performed? If I am not mistaken, a total of 180 tests were performed:

-30 for the main results in Table 1 (5 outcomes * 6 categories of diseases)

-90 tests for age-stratified results (5 outcomes * 6 categories of diseases * 3 categories of age)

-60 tests for sex-stratified results (5 outcomes * 6 categories of diseases * 2 categories of sex)

Was the BH correction brought to the total of 180 tests?

Line 151: Unfortunately the url is broken and source data is not accessible. Could the other provide and updated url and aggregated data for each outcomes for year 2015-2019 as supplementary materials?

Results

Table 1 and supplementary tables

No information is provided about the fluctuation around the mean outcomes reported for 2015-2019. This would help interpret the differences and associated p-values more easily.

Could the authors provide the maximum and minimum values for each outcome for 2015-2019, or a standard deviation?

Line 165: The figures reported in the text do not match those shown in lines 4 and 5 of section 2 in Table 1.

Line 191 and 194: change “ICU admissions” by “proportion of ICU admissions”

Lines 185, 186; 191, 193: The authors make several claims of statistical significance but no mention of confidence interval or p-values are made in the supplementary tables. Although I can assume that asterisks mean a p-value < 0.05, there is no such indication in the supplementary tables. Again, how was Benjamini-Hochberg correction applied?

Figures 1-5

Again, it is important to visualize the fluctuation around the mean for 2015-2019 to assess the extent of the differences reported for 2020. For 2015-2019, could the authors show on the figures the weekly maximum and minimum, or each year’s curve individually?

In 2020, a steep decrease is observed for all natural causes after EW45. In the methods, the authors explain that data beyond EW48 was ignored; could this drop be attributed to a greater delay in reporting data for the end of year 2020 that affected EW45 to EW48 as well?

I find that the text and figures lack a consistent designation for each non-covid specific category; sometimes the ICD chapter is used, sometimes different designations (e.g. “diseases of the circulatory system” in Table 1, “Chapter IX” in Figure 5, “cardiovascular diseases” in Figure 5 legend). The manuscript would benefit from an improved consistency on that matter.

The labeling of figure 3 and 4 is incoherent: the legends refer to neoplasms and respiratory diseases, respectively, whereas the figures themselves refer to chapter X (respiratory diseases) and II (neoplasms), respectively. Given the shape of the mean trend for 2015-2019 (seasonal in figure 3 and constant in figure 4), I guess that the information on the figures and correct and the legends are wrong.

The multiplication of the figures for each specific cause, each with a very similar legend does not help understanding. I suggest simplifying the analysis and the presentation of the results by grouping the non-covid causes of admission in two categories: seasonal or communicable (respiratory and infectious diseases) vs. non-seasonal or non-communicable (cardiovascular diseases and neoplasms), based on the observation that each follow similar patterns as noted in the manuscript.

Line 198: “beginning of the pandemic (EW11-21)” the first red line in the figures is set to EW9 whereas in the methods (line 120) it is mentioned that the first control phase started in EW11.

Line 198-199 “there was a reduction in the number of non-COVID-19 hospitalizations” On the EW11-EW21 there was almost no cases of COVID-19 reported. This statement should mention all causes of admissions.

Line 200-201: “which then rises steeply after the first reopening at EW21”. The increase seems to start earlier for all natural causes (week 15 for hospital admissions, week 17 for ICU and deaths). The rise is only seen in week 20 for COVID19, and is then possibly aggravated by the removal of control measures in week 21. Beyond week 21 we can see clearly a difference between the dynamics of COVID19 and non-COVID19 causes, likely due to a better classification of cases in an epidemic context.

Lines 206-207: I would only consider this statement to hold true beyond EW21.

Discussion

The discussion is interesting and in agreement with the results. However some points would benefit from more caution and may need revision in the light the concerns exposed above.

The practice and the quality diagnosis and ICD classification of patients would merit more discussion, in the light of my preliminary comments.

Could part of the decline of hospitalizations for specific causes could be explained by a coding artifact of admissions: the preferential coding of COVID19 cases over any underlying condition (eg: cardiac failure, respiratory failure, cancer etc.)?

Line 314: I would be more cautious before interpreting the differences between men and women for a single outcome and a specific category of diseases, given the possible limits of the data.

Line 366: It seems that there were still quite some lack of information from EW40, explaining the drop at the end of year 2020 (see my comment in the results

Line 383: This final statement is quite welcome.

6. PLOS authors have the option to publish the peer review history of their article (what does this mean?). If published, this will include your full peer review and any attached files.

**Do you want your identity to be public for this peer review?** For information about this choice, including consent withdrawal, please see our Privacy Policy.

Reviewer #1: No

Reviewer #2: No

---

## [Decision Letter · Decision Letter 1]

22 Nov 2021

The impact of COVID-19 pandemic course in the number and severity of hospitalizations for other natural causes in a large urban center in Brazil

PGPH-D-21-00228R1

Dear Dr. Brant,

We're pleased to inform you that your manuscript has been judged scientifically suitable for publication and will be formally accepted for publication once it meets all outstanding technical requirements.

Within one week, you'll receive an e-mail detailing the required amendments. When these have been addressed, you'll receive a formal acceptance letter and your manuscript will be scheduled for publication.

An invoice for payment will follow shortly after the formal acceptance. To ensure an efficient process, please log into Editorial Manager at https://www.editorialmanager.com/pgph/ click the 'Update My Information' link at the top of the page, and double check that your user information is up-to-date. If you have any billing related questions, please contact our Author Billing department directly at authorbilling@plos.org.

Kind regards,

Mathieu Nacher

Academic Editor

Additional Editor Comments (optional):

Reviewers' comments:

Reviewer's Responses to Questions

**Comments to the Author**

1. If the authors have adequately addressed your comments raised in a previous round of review and you feel that this manuscript is now acceptable for publication, you may indicate that here to bypass the “Comments to the Author” section, enter your conflict of interest statement in the “Confidential to Editor” section, and submit your "Accept" recommendation.

Reviewer #1: All comments have been addressed

Reviewer #2: All comments have been addressed

2. Does this manuscript meet PLOS Global Public Health’s publication criteria? Is the manuscript technically sound, and do the data support the conclusions? The manuscript must describe methodologically and ethically rigorous research with conclusions that are appropriately drawn based on the data presented.

Reviewer #1: Yes

Reviewer #2: Yes

3. Has the statistical analysis been performed appropriately and rigorously?

Reviewer #1: Yes

Reviewer #2: Yes

4. Have the authors made all data underlying the findings in their manuscript fully available (please refer to the Data Availability Statement at the start of the manuscript PDF file)?

Reviewer #1: Yes

Reviewer #2: Yes

5. Is the manuscript presented in an intelligible fashion and written in standard English?

Reviewer #1: Yes

Reviewer #2: Yes

6. Review Comments to the Author

Reviewer #1: The authors have answered all the remarks and reservations in a relevant and well argued manner. The conclusions presented are well supported by the data and analyses performed. The publication of this article does not call for any reservation on my part.

Reviewer #2: Thank you for this revised version of the manuscript and the careful answer to each one of my concerns.

The precisions brought to the text and figures are welcome clarifications.

I find in particular very interesting the figures included in the supplementary material and I appreciate the effort made to explain how to access the data from SUS.

7. PLOS authors have the option to publish the peer review history of their article (what does this mean?). If published, this will include your full peer review and any attached files.

**Do you want your identity to be public for this peer review?** For information about this choice, including consent withdrawal, please see our Privacy Policy.

Reviewer #1: No

Reviewer #2: **Yes: **Yann Lambert
